# Towards The Internet of Smart Clothing: A Review on IoT Wearables and Garments for Creating Intelligent Connected E-Textiles

**Tiago M. Fernández-Caramés** *[ID] and **Paula Fraga-Lamas** *[ID]

Department Computer Engineering, Faculty of Computer Science, Universidade da Coruña, 15071 A Coruña, Spain
* Correspondence: tiago.fernandez@udc.es (T.M.F.-C.); paula.fraga@udc.es (P.F.-L.);
  Tel.: +34-981-16-70-00 (ext. 6088) (T.M.F.-C.)

**Abstract:** Technology has become ubiquitous, it is all around us and is becoming part of us. Together with the rise of the Internet of Things (IoT) paradigm and enabling technologies (e.g., Augmented Reality (AR), Cyber-Physical Systems, Artificial Intelligence (AI), blockchain or edge computing), smart wearables and IoT-based garments can potentially have a lot of influence by harmonizing functionality and the delight created by fashion. Thus, smart clothes look for a balance among fashion, engineering, interaction, user experience, cybersecurity, design and science to reinvent technologies that can anticipate needs and desires. Nowadays, the rapid convergence of textile and electronics is enabling the seamless and massive integration of sensors into textiles and the development of conductive yarn. The potential of smart fabrics, which can communicate with smartphones to process biometric information such as heart rate, temperature, breathing, stress, movement, acceleration, or even hormone levels, promises a new era for retail. This article reviews the main requirements for developing smart IoT-enabled garments and shows smart clothing potential impact on business models in the medium-term. Specifically, a global IoT architecture is proposed, the main types and components of smart IoT wearables and garments are presented, their main requirements are analyzed and some of the most recent smart clothing applications are studied. In this way, this article reviews the past and present of smart garments in order to provide guidelines for the future developers of a network where garments will be connected like other IoT objects: the Internet of Smart Clothing.

**Keywords:** smart clothing; smart garments; wearables; Internet of Things; IoT; e-textiles; electronic textiles; Industry 4.0; biometrics; sensors; blockchain

---

## 1. Introduction

Smart wearables can be defined as electronic devices intended to be located near, on or in the body to provide intelligent services that may be part of a larger smart system thanks to the use of communications interfaces.

Smart clothes can be created by embedding smart wearables into garments. Clothing is the only wearable that adjusts to our everyday lifestyle over the course of a lifetime. It represents a strong candidate to become the next interface between the real and the digital world, replacing or extending smartphones and other portable connected devices. In addition, textiles are essentially the ultimate wearable medium since the physics of fabrics and the power of flexible electronics are a perfect match: form-fitting, adaptable and usually in direct contact with our body. Furthermore, shirts are more natural to wear than a wristband or a chest strap, and are able to track more biometric signals due to the larger area of skin they cover. Thus, biometric tracking becomes easier, since signals are picked up exactly where they happen and, hence, it is possible to maximize their meaning. For example,

readings of heart rate signals from body extremities are very different from readings obtained on the chest. Thus, smart clothes are able to monitor, document, augment and actuate on our lives providing unprecedented opportunities for tackling global societal challenges in areas like smart and crowdsourced health [1], ageing [2], work safety [3] or personal productivity [4].

The Internet of Things (IoT) is already unlocking the benefits of the data economy in numerous industries like healthcare [5], agriculture [6], manufacturing [7], home automation [8–10], transportation [11], energy [12,13], emergency management or defense and public safety [14,15]. Moreover, IoT and wearables, when coupled with advances in 5G communication networks for device-to-device communications [16], virtual/augmented reality [17,18], cyber-physical systems [19], Artificial Intelligence (AI) [20] and smart textiles [21], can bring human-to-human and human-to-machine connectivity and interaction to a new level. Therefore, smart wearables and smart clothing are on the limit that separates the physical and the digital world, and when combined with other technologies (e.g., smart glasses [22]), they have the potential to transform society due to their large-scale use and their transformative effects in many industries [23]. The mentioned fields are paving the way to the novel paradigm of 'Internet of Smart Clothing', which envisions a world where smart garments communicate with each other, with the objects on their environment and with remote servers on the Internet in order to provide advanced services.

According to a 2016 European Commission report [23], there are signs that indicate that wearables and smart garments will be utilized in the near future on a much larger scale. For such applications, the next generation of wearables, fabrics and garments needs to be flexible, fashionable, and ideally, in some cases, invisible. Moreover, they may provide efficient power management, make use of energy harvesting devices and carry out high-performance computing tasks [24].

This article introduces a comprehensive analysis of the evolution of smart wearables and garments, and their main characteristics and applications. Unlike other previous reviews [25], the main contribution of this work focuses on presenting a holistic approach to smart garments with a thorough study on the most relevant challenges. Thus, the aim of this article is to envision the potential contribution of smart garment technologies for revolutionizing industry and people's quality of life through the Internet of Smart Clothing.

The rest of this paper is organized as follows. Section 2 reviews the history of smart wearables and garments, their types, the main end-user requirements and the basic components of a smart garment system. Section 3 presents some of the most promising commercial and academic applications for IoT-enabled smart garments. Section 4 analyzes the opportunities on the smart wearable and clothing market. Section 5 describes the main current technical limitations and outlines the primary challenges that stand in the way of leveraging IoT-based smart wearables and garments technologies at a broader scale. Finally, Section 6 is devoted to conclusions.

## 2. Basics of IoT Smart Wearables and Garments

### 2.1. Overview of Early Wearable Computing

The initial attempts to create smart clothing are related to the first wearable computers. Specifically, the first wearable computer is often attributed to Edward O. Thorp and Claude Shannon [26,27], who conceived in 1955 (and implemented later in 1960–1961) a cigarette pack sized device with only twelve transistors that was used to beat Las Vegas casinos at the roulette wheel (it allowed for timing the revolutions of the ball on a roulette wheel and determine where it would end up). The computing device was worn on the waist and its output consisted of a speaker behind the ear, while the input was provided by a toe-switch.

Also in the 1960s, Ivan Sutherland [28,29] performed the first experiments with smart glasses and helmets, which paved the way for creating the field of Augmented Reality (AR). In fact, such a field was actually not developed until the early 1990s: first empirically by Caudell and Mizell [30], from Boeing,

who presented a head-up see-through display used to augment the visual field of an operator with information related to the task she/he was carrying out, and then theoretically by Azuma [31].

After the initial works in the 1960s, only a few pioneers worked on wearable computing projects. One of them was Steve Mann, who created in the 1970s a wearable system for assisting photographers that characterized the way scenes and objects responded to light [32]. The same author continued to work in multiple wearable computing projects during the 1980s and 1990s, which included a wearable radar system for the blind, audio wearables, AR systems and mediated reality wearables. Note that mediated reality goes a step further than AR in the sense that visual content not only can be added to reality, but it can also modify it or diminish it deliberately so that the perception of a user on the real objects on his/her environment can be altered [33].

At the end of the 1990s it was presented one of the first academic smart clothing platforms, which was developed for the DARPA by the Georgia Institute of Technology: the Wearable motherboard [34]. Such a device was actually a smart shirt aimed at monitoring vital signs in an unobtrusive manner for healthcare and battlefield management applications.

Regarding commercial wearable computers, some authors consider the Mobile Assistant, manufactured by Xybernaut and launched in 1996, as the first commercial wearable computing device [35]. Specifically, the Mobile Assistant provided custom programs and user interfaces to mechanics and technicians that worked for the military and commercial sectors, as well as for healthcare personnel. Unfortunately, the Mobile Assistant was considered to be too bulky and had battery issues, but, nonetheless, its commercialization supposed a breakthrough.

It must be indicated that in the 1970s and 1980s other devices provided portability and wireless communications to certain stand-alone applications (e.g., the first cell phone (1973), the original walkman (1979), the first commercial heart rate monitor for athletes (1983)), but, due to their application and computing resource limitations, such products were just considered as light wearable computing devices. Actual smart clothing prototypes came much later, being one of the first ones the Cyberia survival suit from Reima [36]. Such a suit was presented in 2000 and was aimed at preventing accidents by monitoring the physical condition and position of the user in Artic environments. Thus, it included a GPS receiver and different sensors connected to electrodes embroidered into the fabric (e.g., hydrometer, thermometer).

Although the initial smart clothing prototypes were bulky and lacked certain features required for being commercialized on a massive scale, they have currently been enhanced remarkably thanks to the recent improvements in electronic miniaturization, energy efficiency, connectivity and in the capacity to embed intelligence into electronic (and photonic) systems, as well as the drastic reduction in the price of the components. This paper is not aimed at detailing the evolution of smart wearables and garments, but the interested readers can find good overviews in [37–40].

### 2.2. Types of Smart Wearables

The smart wearables that are embedded into smart clothing can be classified according to diverse parameters. The TC (Technical Commission) 124 of the IEC (International Electrotechnical Commission), which is an ongoing effort to standardize the field of wearable electronic devices and technologies, distinguishes among four different types of smart wearables [41]:

- Accessory wearables. They are low-power devices that are adapted to the human body in order to be worn as accessories like smart watches, smart glasses or fitness trackers.
- Textile/Fabric wearables. They integrate electronics into textiles through flexible fabrics. The European Center for Standardization categorized in 2011 this kind of wearables, while defining them as functional textile systems that interact with its environment (i.e., they adapt or respond to changes in the environment) [42].
- Patchable wearables. They are skin-patchable devices that are flexible and very thin.
- Implantable wearables. They are lightweight self-powered wearables that are implanted into the human body without any health concerns.

Similarly, the IEC Standardization Group (SG) 10 on wearable smart devices [43] indicated that the previously mentioned types of wearables can be classified according to their location near, on or in an organism (e.g., the human body), distinguishing among:

- Near-body wearables. They are intended to be located near the body but they do not need to contact it directly.
- On-body wearables. They are located on the body, in direct contact with the skin.
- In-body wearables. They are implanted inside the body.
- Electronic textiles. They make use of fabric or textile-based electronics and components [44].

In addition, as it is illustrated at the bottom in Figure 1, there are multiple methods of attaching electronics onto textiles [45] that differ mainly on their level of integration into clothing. Thus, such types of e-textiles, ordered from the lowest to the highest level of integration, can be for example: attached with clips or to belts, magnet or velcro-attached, based on flexible electronics, sewn-in e-textiles and smart clothing.

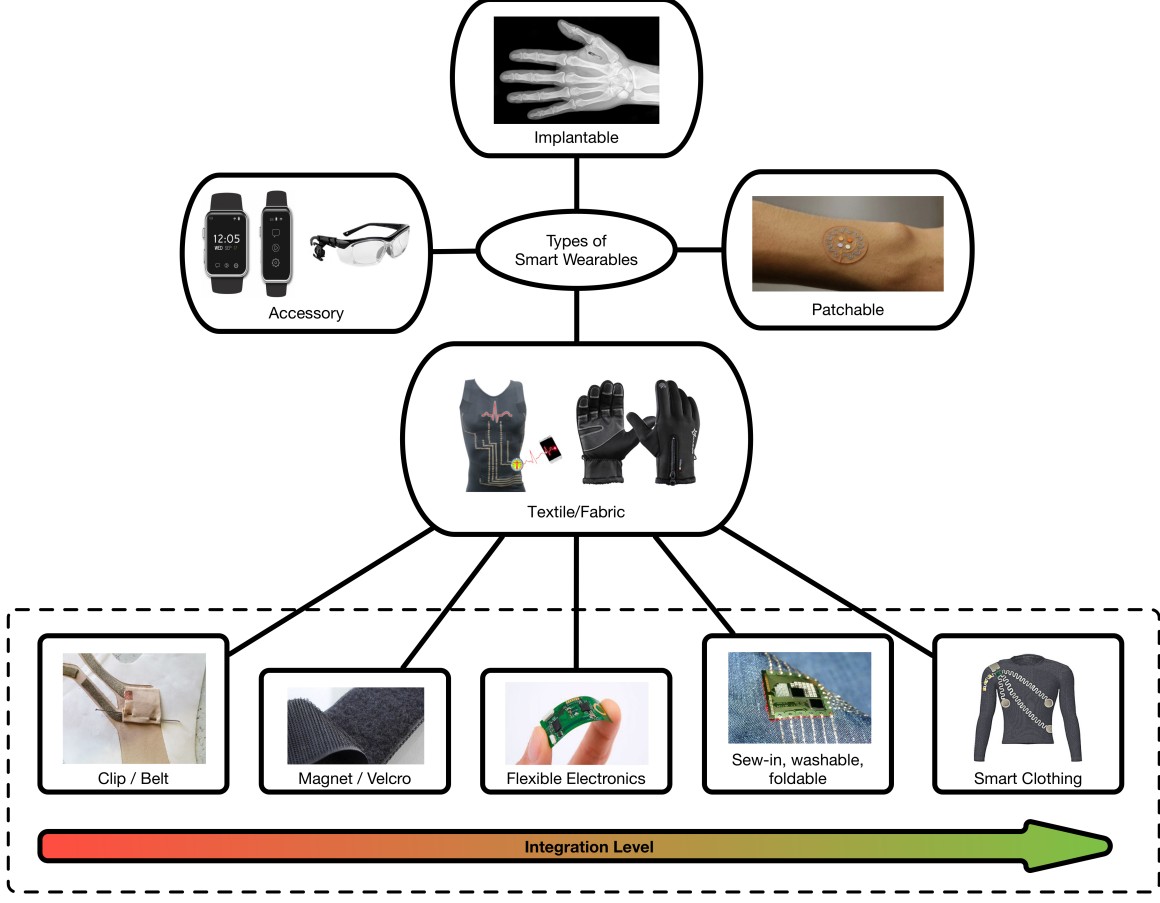

**Figure 1.** Main types of smart wearables and textile/fabric wearables.

All the previously mentioned types of smart wearables may be applied to similar end-user applications, but they require different levels of integration, power and degree of tailoring to the human body. The IEC TC 124 considers that we are currently on the first phase on the evolution of smart wearables, since accessory wearables are already mature and commercially available on a wide scale, but academia and industry are working towards the second phase in order to reach around 2020 the same level of maturity in textile/fabric wearables. Moreover, the use of new materials, designs, energy storage, energy harvesting technologies and production techniques, have provided a strong push factor to improve performance, functionality and usability [46].

Due to this reason, this paper is essentially focused on the study of the textile/fabric wearables or electronic textiles that will be part of the second phase on the smart wearable evolution roadmap of the IEC TC 124 and which are considered as a potential multi-billion market [47]. In addition, smart garments now involve multiple components and diverse industries and disciplines (illustrated in Figure 2), which will be also analyzed throughout this article.

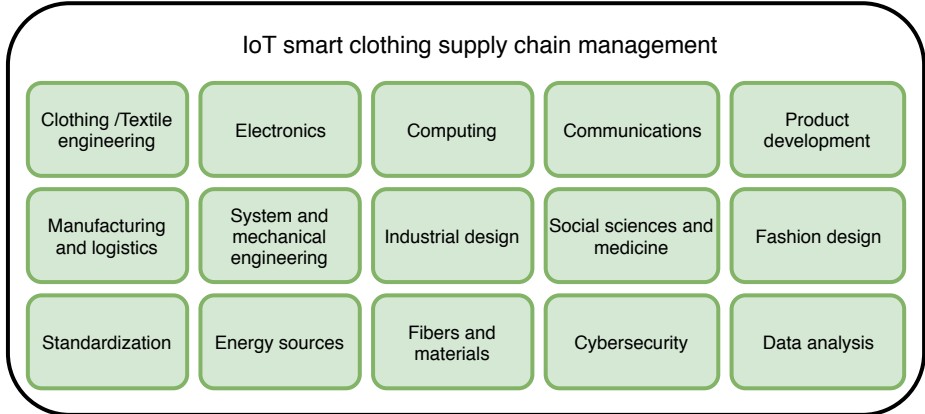

**Figure 2.** Disciplines involved in the IoT smart clothing supply chain management.

### 2.3. End-User Design Requirements

From the end-user point of view, smart clothing has to be designed by taking multiple factors into account [48]:

- Technical requirements. Smart wearables and garments have to be ruggedized to support daily and/or sport activities. In addition, their batteries have to last enough to power the embedded electronics during the activity to be monitored.
- Functionality. Smart clothing has to be comfortable, so it has to be adaptable to the human body. Moreover, clothes that embed electronics must be safe and have to be flexible to adapt to the body movements. It is also really important to take into account the thermal regulation processes of the human body and the potential exposition to corporal fluids (e.g., sweat, blood). More details on these requirements are detailed below.
- Aesthetics. They are essential for the acceptability of smart clothing. Therefore, the design has to take care of both the materials (e.g., fiber type, yarns, fabric performance) and the aesthetic part (e.g., color, cut/fit, trim).
- Cultural requirements. Like in fashion, it is important to distinguish clothing depending on factors like the wearer community and age group. The same product can be acceptable or unacceptable depending on the user culture, traditions or dress code.

Among the previous factors, usability is one of the most important and its requirements are very different from the ones to be fulfilled in other electronics fields. Specifically, the following are the main factors that influence the design depending on the human body [35]:

- Anatomical characteristics. The design of smart clothing has to consider the body measures and the sex of the wearer, since the shape and fit differ noticeably from one user to another. In addition, posture and movement factors are key in order to provide comfort.
- Physiological characteristics. As it was previously mentioned, human fluids and thermal regulation can damage the embedded electronics, so their impact has to be considered during the design stage. Therefore, drinking, urinary outputs, sweating, thermogenesis and heat dissipation/retention have to be taken into account when designing a smart garment.

## 2.4. Components of a Smart Garment of the Internet of Smart Clothing

### 2.4.1. Communications Architecture

A garment that wants to become part of the future Internet of Smart Clothing has to be composed of most of the essential subsystems that are depicted at the bottom of Figure 3. The communications among the different subsystems may be performed either wirelessly or by using conductive fabrics. The former is often more expensive in terms of hardware and requires more power, but removes the need for designing and embedding conductive yarn-based buses into a smart garment. However, the advent of new conductive fabrics and soft/printed electronics will enable seamless and massive integration of sensors into textiles [49,50]. In any case, note that such an interconnection subsystem is essential for the reliability, washability, wearability and miniaturization of a smart textile, so its selection its key. The reader interested in this topic can find in [51] a detailed review about the different types of conductive fabrics/inks/materials and their fabrication techniques.

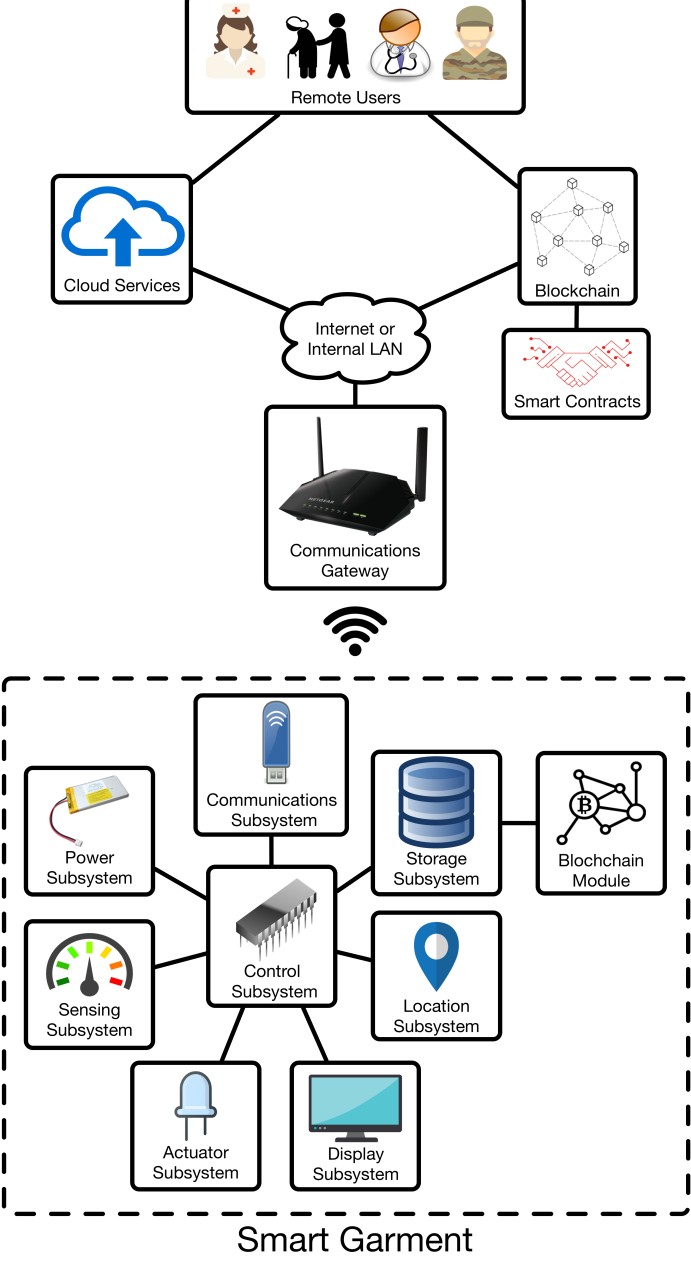

**Figure 3.** Generic architecture of an IoT smart garment system.

Figure 3 also illustrates a typical IoT architecture that supports collecting smart clothing data and that is composed by the following main components:

- A communications gateway that exchanges information with the smart garments in order to send it through the Internet or an internal LAN to remote services provided, for instance, by cloud servers or a blockchain [52]. The communications gateway can also process the received data and provide fast responses to the smart garments, thus acting as an edge or fog computing gateway [53].
- A cloud server that collects and stores data, and provides certain remote services to the smart garments and to remote users (e.g., doctors or nurses that need to access the stored information).
- A blockchain. Although it is not essential for the basic functioning of a smart clothing system, it enables different useful features like redundancy, data security and trustworthiness [52]. Moreover, a blockchain can run smart contracts (pieces of software that translate legal terms into code that can be run autonomously on a blockchain) [54], which allow for automating certain tasks according to the detected events.

It is also important to note that most architectures divide the previously mentioned components into three layers [55]:

- Body Area Network (BAN). The components of each smart garment are connected through a common network topology characterized by providing a really short range (just enough to cover a human body). Actually, since the components are distributed through the human body but embedded in garments, the network is called Wearable BAN (WBAN), in contrast to other cases when such components are inside the body conforming an in vivo or Implantable BAN (IBAN). WBANs need to be energy-efficient, since smart garments mostly rely on batteries [56].
- Personal Area Network (PAN) or Local Area Network (LAN). This network collects data from smart garments and sends them to a cloud or remote server. PANs usually provide shorter ranges than LANs (usually up to 10 m). In the case of smart garments, communications are performed wirelessly, so the terms Wireless PANs (WPANs) and Wireless LANs (WLANs) are often used. An example of WPAN is Bluetooth, while WiFi is a type of WLAN. It is also worth pointing out that at this communications layer it is possible to provide mesh network communications so that smart garments can communicate with each other and with the objects and machines that surround them.
- Wide Area Network (WAN). This is a type of network like the Internet, which covers a really wide area thanks to the support of a distributed infrastructure. It is essential for many IoT applications, but in some cases (e.g., critical infrastructures [19,57] or industrial environments [58]) their services may be provided through an internal LAN.

Figure 3 depicts the main subsystems that take part of a smart garment. The next subsections detail the main components and characteristics of such subsystems.

### 2.4.2. Sensing Subsystem

The sensor subsystem may be composed by several types of sensors that can monitor diverse phenomena or events from the surrounding environment [35,51,59,60]. According to such factors, in smart garments, it can be mainly distinguished among:

- Motion, gesture and position sensors. The most commonly used sensors are accelerometers and gyroscopes, although certain applications make use of a barometer to obtain altitude. Sensors based on infrared or ultrasound sensors are often used to determine proximity [61]. Moreover, Passive Infrared (PIR) sensors can be embedded to detect the movement of people or animals around the wearer. Other motion sensors are tilt-switches, vibration sensors or pedometers.
- Body temperature. There are different sensors that can be used to measure body temperature, like thermistors or Resistance Temperature Detectors (RTDs), although some authors suggested using other new technologies [62–64].

- Vital sign rates. There are sensors for monitoring heart rate, respiration rate, blood pressure, blood leakage, pulse oxygenation, glucose levels, galvanic skin response or electrodermal activity. It is also possible to use embedded sensors to obtain electrocardiograms (ECGs) and electroencephalographies (EEGs) [65–70].
- Location sensors. The devices that can be used for positioning a smart garment are later described in Section 2.4.6.
- Interaction. They mainly detect touch through mechanical switches or switch-tactile sensors, but it is also possible to use capacitive or resistive touch screens. There also exist textile switches, fabric/laser keyboards and even 2D touchpads [71].
- Environmental sensors. They collect information on environmental parameters. Therefore, this kind of sensors measure air temperature, altitude, light (e.g., Light-Dependent Resistors (LDRs), photodiodes), Ultraviolet (UV) light, sound/noise (e.g., microphones, speech recognition sensors), atmospheric pressure, humidity, the presence of certain gases (e.g., $CO$ or $CO_2$), or the presence of Chemical, Biological, Radiological, Nuclear and Explosive (CBRNE) substances [72].
- Surrounding objects. Complimentary Metal-Oxide Semiconductor (CMOS), Charge-Coupled Device (CCD) and infrared cameras can be used to recognize the objects that surround the wearer. Similarly, Radio-Frequency IDentification (RFID) and Near Field Communication (NFC) tags attached to objects can be read from a certain distance through a reader embedded into a smart garment [73].

### 2.4.3. Actuation Subsystem

Actuators allow wearers and the embedded systems to perform certain actions on a smart garment (i.e., interact with it). The following actuators are some of the most popular in smart garments [35]:

- Visual indicators [74]. They show light and image information through Light-Emitting Diodes (LEDs), optical fiber or displays (e.g., Liquid-Crystal Displays (LCDs) or electronic-ink displays (e-ink displays)).
- Sound [75]. They emit sound and voice through buzzers, loudspeakers, headphones, in-earphones or speech synthesizers. In addition, they can also transmit sound above the human hearing range by using ultrasound actuators.
- Movement and vibration [76]. They transform electricity into some sort of movement. Examples of such actuators are electric motors, vibration motors, solenoids or electric valves.
- Heating and cooling [37]. Actuators like resistive heaters can generate heat from electricity for the smart garment wearer, while thermoelectric materials can help in cooling.

### 2.4.4. Control Subsystem

The control subsystem can be implemented by using different types of electronic devices. Examples of such devices are Central Processing Units (CPUs), microcontrollers, Field-Programmable Gate Arrays (FPGAs), Application-Specific Integrated Circuits (ASIC) or System-on-Chips (SOCs). In the case of CPUs, they are computationally powerful but consume too much energy for small battery-powered devices. Microcontrollers are often used for low-power systems because they consume less energy, they can be reprogrammed easily and they have sufficient processing power for carrying out the tasks to be performed. FPGAs provide good performance for executing certain deterministic demanding tasks and can be reconfigured easily with a different design, but they have two drawbacks:

- FPGA design development is usually not as straightforward as microcontroller programming, since it often involves very precise resource (e.g., logic gates, embedded memory) interconnection and synchronization.
- FPGAs usually consume more power than other devices due to their need for powering the used logic continuously.

In the case of ASICs, they are designed explicitly for specific applications and thus they can perform tasks faster than the other mentioned embedded devices. In addition, since they are optimized for specific applications, their power consumption can also be minimized. However, the cost of developing an ASIC is really high (millions of U.S. dollars), so its use only compensates when a really high amount of devices is going to be produced. Regarding SoCs, they integrate into a single circuit a powerful microcontroller and several peripherals (e.g., wireless transceivers), which makes them consume more power than traditional microcontrollers.

Due to the previously mentioned reasons, microcontrollers are almost always the foundations of Do-It-Yourself (DIY) sewable smart clothing platforms, which are the easiest way to develop a control subsystem for a smart garment prototype. The most popular wearable platforms are compared in Table 1. In such a Table it can be observed that, except for the Adafruit Gemma M0, the computational resources of the platforms are quite limited. Nonetheless, since most platforms are currently being used for performing very specific tasks, the selected microcontrollers are computationally powerful enough and have the advantage of consuming much less energy than other hardware. Table 1 also shows that, despite the lack of computational power, the compared platforms are able to use a wide array of accessories, so they can be used in many different smart clothing applications.

### 2.4.5. Communications Subsystem

This subsystem is essential for a smart garment of the Internet of Smart Clothing, since the connection to the Internet or to an internal LAN opens up a wide variety of use cases and applications that can make use of large databases and remote processing power.

It is first worth pointing out that the proposed communications subsystem is responsible for performing two main tasks: to enable data exchanges between a smart garment and a communications gateway, and to provide an identification mechanism for the wearer. Both tasks are considered under the same subsystem because most communications transceivers provide a way to identify the user (e.g., through the MAC address or through an IP and a port).

However, the communications subsystem may be split into two dedicated physical transceivers: one for identification and another exclusive for communications. In this case, different technologies can be used to provide an identifier, like RFID, which is arguably one of the most popular identification systems [77,78]. Such a technology uses radio frequency transponders (usually called tags) that emit unique identifiers and, in some cases, certain information. In addition, an RFID system makes use of an RFID reader that may communicate with the tags through different communications techniques [79]. The most relevant advantage of RFID over other traditional identification technologies (e.g., barcodes, QR codes) is the fact that line-of-sight is not required between a tag and a reader, so it is easier to perform the identification. Moreover, certain RFID tags called passive tags, at short distances (up to several meters), do not need batteries. Nonetheless, there are RFID tags that make use of batteries (active tags) to achieve a longer reading range (usually, up to 100–150 m in real scenarios) [80].

Short-distance readings can also be performed through NFC [81], which can be considered as an improved version of traditional passive RFID tags. However, when longer reading distances are required, certain communications technologies can be re-purposed to provide, at the same time, identification and a communications channel. Examples of such technologies are Bluetooth Low Energy (BLE) or WiFi, which have been demonstrated to be useful in diverse identification and tracking applications [82–84] and can easily reach between 50 and 100 m in real-scenarios.

There are many other identification and communications technologies that can be used by smart garments, like 3G/4G/5G, ultrasounds, infrared, ZigBee [85], Long-Range Wide Area Network (LoRaWAN) [86,87], Dash7 [88], Ultra-Wide Band (UWB), WirelessHART [89], SigFox [90], ANT+ [91], Weightless-P [92], Wi-SUN [93] or IEEE 802.11ah, among others. Table 2 shows a comparison on the main characteristics of the latest and most popular communications and identification technologies that could be used for smart clothing, indicating their frequency band, usual maximum range, data rate, power consumption and relevant features.

**Table 1.** Most popular DIY platforms for developing smart wearables and smart clothes.

| Characteristic \Platform | Lilypad Arduino | Lilypad Arduino Simple Board | Lilypad Arduino USB | Adafruit Flora v3 | Adafruit Gemma v2 | Adafruit Gemma M0 | SquareWear v2.4 | SquareWear Mini | Igloo |
|---|---|---|---|---|---|---|---|---|---|
| Microcontroller | ATmega168V or ATmega328V | ATmega328V | ATmega32U4 | ATmega32U4 | Attiny85 | ATSAMD21E18 (32-bit Cortex M0+) | ATmega328 | ATmega328 | PICAXE 14M2 |
| Clock Rate | 8 MHz | 8 MHz | 8 MHz | 8 MHz | 8 MHz | 48 MHz | 8 MHz | 12 MHz | 32 MHz |
| Flash Memory | 16 KB | 16 KB | 32 KB | 32 KB | 8 KB | 256 KB | 16 KB | 16 KB | 2 KB |
| SRAM | 1 KB | 1 KB | 2.5 KB | 2.5 KB | 512 Bytes | 32 KB | 1 KB | 1 KB | 512 Bytes |
| EEPROM | 512 Bytes | 512 Bytes | 1 KB | 1 KB | 512 Bytes | n/a | 512 Bytes | 512 Bytes | n/a |
| Operating Voltage | 2.7–5.5 V | 2.7–5.5 V | 3.3 V | 3.3 V | 3.3 V | 3.3 V | 3.3 V | 3.3 V | 3–5 V |
| I/O Pins | 14 digital I/O pins, 6 analog inputs | 9 digital I/O pins, 4 analog inputs | 9 digital I/O pins, 4 analog inputs | 8 digital I/O pins, 4 analog inputs (used for the SPI and serial bus) | 3 available I/O pins | 12 analog/digital pins | 8 digital pins, 4 analog pins | 8 digital pins, 4 analog pins | 6 I/O pins (four have a built-in ADC) |
| Size | Diameter: 50 mm, Thickness: 8 mm | Diameter: 50 mm, Thickness: 8 mm | Diameter: 50 mm, Thickness: 8 mm | Diameter: 45 mm, Thickness: 8 mm | Diameter: 28 mm, Thickness: 7 mm | Diameter: 28 mm, Thickness: 6.4 mm | 43 mm × 43 mm | 33 mm × 43 mm | 40 mm × 40 mm × 7 mm |
| Approx. Weight | 4.54 g | 4.54 g | 4.54 g | 4.4 g | 3.29 g | 2.1 g | 11 g | n/a | n/a |
| Accessories | RGB leds, buzzer, vibration motor, light sensor, reed switch, accelerometer, temperature sensor | | | Accelerometer, magnetometer, GPS, RGD leds, a Bluetooth Low Energy (BLE) transceiver, a light sensor, a color sensor, UV light sensor | RGB leds, vibration motor, photocell, tactile switch | | Built-in accessories: RGB led, buzzer, light sensor, temperature sensor | Built-in accessories: mini-buzzer, light sensor, temperature sensor, 16 KB EEPROM, push button | Multiple LEDs, light sensor, buzzer, different switches |
| Price | US$15.95 | US$19.95 | US$24.95 | US$14.95 | US$9.95 | US$9.95 | US$19.99 | US$17.99 | GBP 7.14 |

**Table 2.** Main characteristics of the most relevant communications and identification technologies for smart clothing.

| Technology | Frequency Band | Maximum Range | Data Rate | Power/Main Features |
|---|---|---|---|---|
| ANT+ | 2.4 GHz | 30 m | 20 kbit/s | Ultra-low power, up to 65,533 nodes |
| Bluetooth 5 LE | 2.4 GHz | <400 m | 1360 kbit/s | Low power and rechargeable (days to weeks) |
| DASH7/ISO 18000-7 | 315–915 MHz | <10 km | 27.8 kbit/s | Very low power, alkaline batteries last months to years |
| HF RFID | 3–30 MHz (13.56 MHz) | a few meters | <640 kbit/s | NLOS, low cost |
| Infrared (IrDA) | 300 GHz to 430 THz | a few meters | 2.4 kbit/s–1 Gbit/s | Security, high-speed |
| IQRF | 868 MHz | hundreds of meters | 100 kbit/s | Low power and long range |
| LF RFID | 30–300 KHz (125 KHz) | <10 cm | <640 kbit/s | NLOS, durability, low cost |
| NB-IoT | LTE in-band, guard-band | <35 km | <250 kbit/s | Low power and wide area |
| NFC | 13.56 MHz | <20 cm | 424 kbit/s | Low cost, no power |
| LoRa | 2.4 GHz | kilometers | 0.25–50 kbit/s | Long battery life and range |
| SigFox | 868–902 MHz | 50 km | 100 kbit/s | Global cellular network |
| UHF RFID | 30 MHz–3 GHz | tens of meters | <640 kbit/s | NLOS, durability, low cost |
| Ultrasounds | >20 kHz (2–10 MHz) | <10 m | 250 kbit/s | Based on sound wave propagation |
| UWB/IEEE 802.15.3a | 3.1 to 10.6 GHz | <10 m | >110 Mbit/s | Low power, rechargeable (hours to days) |
| Weightless-P | License-exempt sub-GHz | 15 km | 100 kbit/s | Low power |
| Wi-Fi (IEEE 802.11b/g/n/ac) | 2.4–5 GHz | <150 m | up to 433 Mbit/s (one stream) | High power, rechargeable (hours) |
| Wi-Fi HaLow/IEEE 802.11ah | 868–915 MHz | <1 km | 100 Kbit/s per channel | Low power |
| WirelessHART | 2.4 GHz | <10 m | 250 kbit/s | HART protocol |
| Wi-Sun/IEEE 802.15.4g | <2.4 GHz | 1000 m | 50 kbit/s–1 Mbit/s | Field area networking, Home area networking |
| ZigBee | 868–915 MHz, 2.4 GHz | <100 m | 20–250 kbit/s | Very low power (batteries last months to years), up to 65,536 nodes |

### 2.4.6. Location Subsystem

The location subsystem allows for positioning and tracking the wearer of a smart garment. Such tasks are nowadays straightforward outdoors, since the evolution of Global Navigation Satellite Systems (GNSS), Advanced-GNSS (A-GNSS) and Differential-GNSS (D-GNSS) technologies (e.g., Global Positioning System (GPS), Global'naya Navigatsionnaya Sputnikovaya Sistema (GLONASS) or Galileo) provide an accuracy that is enough for most applications. Nonetheless, positioning a user indoors is more difficult [94], so it is required to study the scenario and make use of specific location techniques that can be applied to most of the communications technologies cited in the previous subsection. Among such techniques, the ones based on Received Signal Strength Indicator (RSSI) or Received Signal Strength (RSS) can provide good accuracy when positioning in limited areas [95–97], but their heavily depend on the scenario and on the used hardware [98]. To avoid such problems, other positioning techniques make use of the Angle of Arrival (AoA) of the received signals [99] or their time of arrival (through Time of Arrival (ToA) and Time Difference of Arrival (TDoA) techniques). These latter techniques work pretty well, but they require really accurate internal clocks and, in the case of ToA, time synchronization between the clocks of the transmitter and the receiver [100].

There are other technologies that enable positioning a smart garment, like traditional cameras, infrared systems [101], ultrasounds [102], UWB [103] or inertial navigation systems [104].

### 2.4.7. Power Subsystem

Powering the electronics embedded into a smart garment is essential for e-textiles. Regarding energy storage devices, batteries are accepted as one of the most important and efficient ways of establishing electricity networks, but other technologies can be used, such as supercapacitors, which have been greatly improved over the last years, reaching energy density levels comparable to lead-acid batteries. In addition, supercapacitors are environmental friendly and offer a high-power density, fast charging/discharging speed and a long life-cycle [105].

Depending on the wearable energy consumption, the requirements of the batteries change significantly. Thus, three different types of devices can be distinguished depending on their power needs [35]:

- Low-end devices. They are devices like traditional watches or pacemakers that can be powered for a long time (years) with button-type batteries due to their really low energy consumption (usually under 100 µW).
- Mid-range devices. They consume an average of 500 mW mainly due to the use of wireless communications transceivers. They usually last less than a day (often just several hours) transmitting continuously, although some technologies make use of sleep modes or periodic transmissions (e.g., BLE beacons) to last much longer. Therefore, this kind of devices require bulkier batteries than low-end devices (e.g., AA or AAA batteries), what makes them currently less appropriate to be embedded into smart clothing.
- High-end devices. These are devices like smartphones or laptops than consume up to 50 Watts. They usually make use of Li-ion batteries, which can be bulky and add significant weight to a garment.

Table 3 includes examples of wearable devices, their power consumption and the basic characteristics of the battery they usually carry. The operating period of a battery is determined by the power requirements and the energy density of the batteries employed. The energy density is the amount of electrical energy stored in a battery of a given weight (Wh/kg). It must be noted that the indicated values just represent examples and may change depending on the use case and on the specific model and manufacturer.

**Table 3.** Examples of wearable devices and the basic characteristics of their batteries. (*) The operating period depends on the usage frequency.

| Device | Power | Battery Type | Operating Period | Weight (g) | Size (l × w × h/d × h, mm) |
|---|---|---|---|---|---|
| Watches | 3–10 µW | Silver oxide button | 1–2 years | 2.4 | 11.6 × 5.4 |
| Pacemakers | 25–80 µW | Lithium button | 7–10 years | 2.83 | 20 × 3.2 |
| Hearing aids | N/A | Zinc-mercury oxide | 25–30 days | 0.3–1.85 | (5.8–11.6) × (3.6–5.4) |
| Digital clocks | 13 mW | Silver oxide button | 6–10 months | | |
| LEDs | 25–100 mW | Silver oxide button | 6–12 months * | 15–25 | (5.8–11.8) × (1.65–5.4) |
| Pedometers | 250 mW | Silver oxide button | 1–2 years * | | |
| Portable radio | 500 mW | AAA | 3–6 months * | 8.5–11 | 10.5 × 44.5 |
| RF circuits | 300–800 mW | AA | <10 h | 14–31 | 14.5 × 50.5 |
| Smartphones | 4–10 W | Li-ion rechargeable | days | 45 | 58.42 × 43.18 × 5.08 |
| Laptops | 50–80 W | Li-ion rechargeable | 3–10 h | 350 | 248.92 × 91.44 × 48.26 |

In the past decades, considerable efforts have been made to face bulky, rigid, and non-flexible batteries, improving their flexibility without sacrificing their performance to integrate them in smart textiles [35]. Researchers have created full cells using conventional printing-based methods and provided protocols for designing inks [106]. Nevertheless, for high-power applications, current battery technologies (e.g., lead-acid, alkaline, NiCd, Li-ion, Li-polymer) are a bottleneck when designing and implementing smart clothing, what leaves two options: to reduce electronics power consumption or to make use of other power sources. Among the latter, it seems that, due to the weight and size requirements, certain energy harvesting systems may be a good fit for many smart garments.

Energy harvesting devices are able to convert diverse types of ambient energy into electricity that can be stored in batteries. There are two main energy sources that may be harvested by smart garments from the human body: body motion and heat. Body motion generates mechanical energy that can be harvested through thermoelectric systems, while the heat dissipated by the body generates thermal energy that can be collected through, for instance, piezoelectric devices.

Besides the energy generated by the body, other power sources exist in the environment, like the sun (collected by photovoltaic cells), the wind (collected by wind mills) or the electromagnetic waves that surround us (which can be rectified through specific energy harvesting devices).

Another important aspect for the power subsystem is how to recharge a smart garment. USB connections or DC-power jacks can be waterproof and embedded into clothing, but there is also the possibility of recharging the batteries wirelessly. Wireless chargers are already available in the market for smartphones, but they are usually quite inefficient and require the wireless receiver to be next to the transmitter [107,108].

The interested reader can find a good overview on the topic in [35].

### 2.4.8. Storage Subsystem

The data collected by a smart garment are usually first processed lightly in the control subsystem (e.g., unit conversion, filtering, or statistical processing) and then stored locally in static memories such as EEPROMs or SD cards [109]. However, in IoT devices, it is common to upload information to an external server [110]. For instance, it is common to upload the collected information to a cloud server or to a fog computing gateway [111,112] that further processes and stores the data, which can ultimately be presented to the user through a graphical interface, (e.g., through a web application).

Smart garments are hardware-based, but they can include a software-based back-end. In fact, this is where the value is created. The connection with the back-end is frequently performed via smartphones [113], although, in the Internet of Smart Clothing, a smart garment would be connected to the Internet or to an Internal LAN through a wireless interface that enables interacting with it.

However, although local storage can be inexpensive, it can be exposed to technical problems (e.g., disk failures) and external factors (e.g., impacts from external objects), which may harm and derive into the loss of the stored data. Remote storage systems (e.g., cloud-based storage systems,

Network-Attached Storage Systems (NAS)) provide remote access and often redundancy to the stored data, but their management usually depends on external companies that may be exposed to Denial of Service (DoS) attacks, which affect the availability of the service, and that in many cases are not be able to guarantee the integrity and trustworthiness of the data. Due to such limitations, in the last years, blockchain has arisen as a new way of storing data (or proof of such data) so that they can be transferred in a secure way and among entities that do not trust each other. Blockchain can be considered to be still under development in many aspects [52,114], but some of its applications for fields where trust is a necessity to interact with third-parties have been already studied [115–119].

Besides interacting with a blockchain, a smart garment can also take part in smart contracts. A smart contract can be defined as a self-sufficient decentralized code that is executed autonomously when certain conditions of a business process are met. Thus, the code translates into legal terms the control over physical or digital objects through an executable program. Smart contract conditions are based on data extracted from the real world by external services that store them into the blockchain (or vice versa). Such services are called oracles. For example, an oracle can inspect records to identify whether an asset has arrived to a warehouse and it can write the arrival information on the blockchain. Then, the smart contract might trigger a conditional statement based on the read value and execute a block of code. Due to the expected impact of blockchain and smart contracts on future IoT devices [52,54], Figure 3 shows a specific blockchain module associated to the storage subsystem.

### 2.4.9. Display Subsystem

Since an IoT smart garment is assumed to be connected to remote storage systems and to provide connectivity to mobile devices, it does not necessarily need to embed a display, but it may be a good complement to avoid using external devices when interacting with the smart garment. Nonetheless, it is worth noting that a display increases significantly energy consumption and that they may have problems with low and high temperatures, drops, pressure or external impacts.

Display technologies like Liquid-Crystal Display (LCD) and its variants (e.g., Twisted Nematic (TN), Super-Twisted Nematic (STN), Vertical Alignment (VA) or In-Plane Switching (IPS)) may be embedded into smart wearables and garments thanks to its recent reduction in price, but, except in very specific cases [120], they need to be refreshed roughly 30 times per second and require the use of a backlight. These facts imply that, in general, LCD technology is not the best fit for smart garments whose battery life is important.

Organic Light-Emitting Display (OLED) technology and its variants (e.g., Active Matrix OLED (AMOLED) or Passive Matrix OLED (PMOLED)) have gained a lot popularity in the last years due to their color definition, fast response time, wide viewing angles, high brightness/contrast and the fact that they consume 20–80% less power than an LCD of the same size [121]. However, OLED displays are still expensive.

Electronic-ink (e-ink) displays only consume power during the process of updating the shown information (i.e., no power is consumed when the information is displayed), so battery life can be extended remarkably [122]. However, e-ink color displays are currently expensive, so most of the ones available in commercial products only shown grayscale images. In addition, e-ink displays have no backlight, so they require ambient light to read them, what may be a problem in certain scenarios (e.g., at night or in industrial specific low-light scenarios).

The future screens for smart garments seem to be related to the emerging field of flexible electronics, which enable printing foldable displays on paper or plastic substrates. Although the latest systems still have certain performance limitations, they have already been embedded in smart devices [123]. There are other emerging technologies based on optical fiber [124], electrochromic [125] and paper-like displays [126], but further research is still required to optimize its integration into smart clothes.

### 3. Smart Clothing Applications

As it can be seen in Figure 4, nowadays wearables offer a number of opportunities in multiple fields in order to improve human life using data-driven customized services. As a result of the embedded intelligence, seamless connectivity and an ever-increasing usability, wearables offer opportunities for activity and condition monitoring, decision-support, actuation, location applications, identification, personal contextual notifications, event detection, information (video/image/audio) display and virtual assistance.

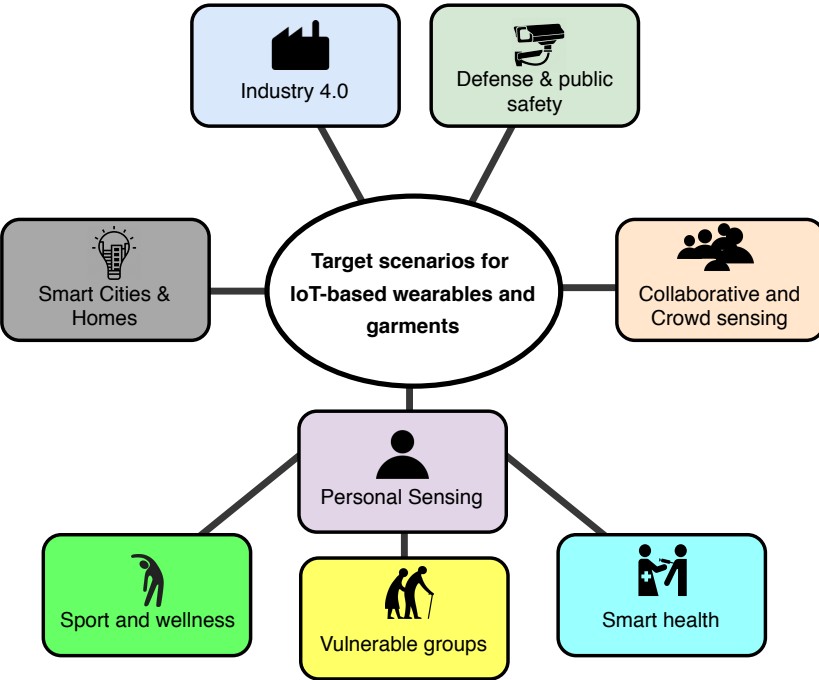

**Figure 4.** Promising target scenarios for IoT-based wearables and garments.

In the next subsections the most relevant commercial applications and academic developments are analyzed to give a global overview on the state of the art of smart clothing and the current context. It must be noted that, although in the last years there has been a lot of research on the field of smart wearables and e-textiles [51,127–129], only a few publications have suggested the integration of smart clothing into the Internet of Things. Therefore, in the next subsections, for the sake of fairness, there will be given examples of both IoT-enabled smart clothing and non-IoT enabled garments with communications capabilities that might make them able to be integrated into the future Internet of Smart Clothing.

#### 3.1. Main Commercial Applications

Currently, the most popular commercial wearables are fitness trackers and smart watches [130]. Nonetheless, the list of devices being introduced to the market or that are yet to come, is much larger, including smart patches, headsets, exoskeletons (i.e., wearable robots [131]), as well as smart jewelry (e.g., a hearing-aid designed in the form of ear rings and smart jewelry to monitor sleep quality [132–134]). It must be emphasized that most of the current commercial initiatives related to wearables and smart clothing cannot be consider IoT-enabled and, to the knowledge of the authors, none of them uses a blockchain or any other Distributed Ledger Technology (DLT) to receive, validate, store and share the collected data with the objective of avoiding untrusted sources.

The following are examples of the main applications:

- Garments and/or wearables to discover and challenge a user physical boundaries, uncovering his/her inner athlete and looking for outperforming his/her own goals [135–138]. It can even be

tracked, monitored and managed the user's emotive and physical state for injury prevention [139] or self-improvement (how to operate at peak efficiency, recommending for example when to slow down, speed up or take a break).

- Garments and/or wearables to help fitness coaches to develop smarter and more targeted programs based around biometrics and spatio-metrics reducing the risks of injury and optimizing rehabilitation [140–146]. Also, collaborative learning with other athletes can be offered.
- Garments and/or wearables to help blind and visually impaired people to become more mobile and independent [147].
- Garments and/or wearables to lose weight [148] or help nutritionists to provide advice and meal plans according to metabolic patterns data.
- Garments and/or wearables to help chronically ill patients (e.g., with epilepsy, diabetes, cardiovascular illness, Parkinson [149]) to make changes in their lifestyle. For instance, such devices may ease the diagnosis of epilepsy and seizure syndrome [150,151], and the effective management of the condition to provide the best possible life quality without stigmas. In addition, a smart band might tell diabetics when their blood sugar is running low [152]. Another application is the detection of asthma through acoustical monitoring of wheeze (one of the major symptoms of an asthma attack) [153].
- Wearables to help to optimize professional growth giving insight into emotions (e.g., level of stress) during the day [154–158]. This information helps to modify the performed actions or understand if and why someone is feeling tense [159].
- Biometrics to help people with jobs that put them in danger, such as police or fire-fighters. For instance, there are smart garments and/or wearables to prevent work-related injuries (e.g., heat stress for firemen [160]).
- Clothes made with an ink that can detect changes in air quality (or even flatulence [161]), heat, moisture, and UV light may switch colors depending on the environment. Particularly, in big cities where pollution is an issue, the user can have a monitor on his/her external clothing to let him/her know if he/she is in an area where there might be chemicals in the air or heavy pollutants above a certain level.
- UV alerts in tiny sensors [162] and/or swimwear and beachwear [163] that determine if the wearer needs more sunscreen.
- Pregnancy biometrics monitoring [164] and preventive devices for postnatal depression [165].
- Baby monitors [166,167] that allow parents to obtain the child's location and actions (for instance, by using a thermometer, a GPS, cameras or proximity sensors).
- Sleep monitors that information about the relationship between active and rest time [168].
- GPS tracking and 'show you the way' devices embedded into shoes or socks [169,170].
- Wearables to help elderly care [171]. They can be sewn to a sock or a slipper to evaluate a person's balance or whether the user is at risk of a fall (or whether the wearer has already fallen).
- Wearables to provide alerts [172]. For example, remote parents can be warned if a cold epidemic outbreaks at their children's school.
- Together with IoT devices, garments can unlock doors, authenticate transactions, identify users or control/actuate on things (e.g., Internet of Medical Things (IoMT) [173]).
- Smart connected garments [174,175] and wearables [176] that provide remote emotional haptic feedback, like sending intimacy (a hug) from remote locations.
- Companies that offer a catalogue of textile solutions and integrated systems: Myant [177], AiQ [178], Interactive wear [179], Myclim8 [180], TexRay [181] or FENC [182].
- Artistic and fashion-designed projects [183–185] that are able to change the color/odor/transparency of the clothes according to the user mood or the interaction, thus making it possible to alter clothing at will (i.e., grey for the office and red for cocktails in the evening) or help to straighten the posture [186] and run and walk so that to protect back and knees.
- Garments and/or wearables to detect breast cancer like the medical project Itbra [187].

Besides the previously mentioned commercial developments, the next subsections analyze the academic systems presented in the last years in different fields.

*3.2. Academic Developments*

3.2.1. Smart Health

Biometrics, like electrocardiogram (ECG), electromyogram (EMG), electroencephalography (EEG), physical stress levels, breathing patterns, sweat, saliva temperature or calorie/energy burnt rate, are common variables that originate from bio-signals. In addition, other vital signs, like psycho-emotional data, can indicate emotions or predispositions derived from specific biological patterns and, sometimes, from multiple signals.

These signals have high intrinsic value to the user as a health enabler. Live bio-signals acquired organically, continuously, passively and dynamically through smart garments can become very useful. Contrary to markers such as genetic information or blood samples, bio-signals are dynamic and continuous, and can be used for real-time analysis, prediction and pattern recognition. Biometrics also opens up the realm of emotions, a whole new dimension of content for social applications (e.g., sharing mood-related state on Facebook, interpreting emotions) or they can even be monetized as e-commerce experiences (e.g., customized recommendations based on product reviews or buying predisposition based on biometrics).

The Internet of Smart Clothing can contribute significantly to tackle societal challenges while reducing the rising economic burden of healthcare services. Physiological remote monitoring can be used to help prevent and diagnose health problems at an early stage, moving from treatment to prevention approaches (e.g., sleep disorders [188]). For instance, in [189] the authors present an IoT-enabled system to prevent obesity that collects sensor data through a smartphone that acts as gateway. Similar communications architectures are proposed in [190,191]. Specifically, in [190] the authors present a T-shirt to monitor the ECG signals from the wearer through a BLE connection that can be read by a smartphone, which might upload them later to a remote server. In [191] the same communications architecture is proposed for collecting information from a smart clothing prototype aimed at monitoring the ECG and the respiration of a patient. In the case of the prototype described in [192], a smart garment is evaluated for monitoring the respiration rate, providing wireless connectivity through a Wireless Sensor Networks (WSN) mote that is cable to carry out mesh network communications (however, the system was not explicitly conceived for the IoT).

Besides heart rate, respiration rate and ECGs, other vital signs can be monitored in a similar way through smart garments, like blood pressure, pulse rate, respiration effectiveness, blood oxygen saturation (SpO2), blood glucose, skin perspiration, body temperature or patient motion. There is a fair amount of literature devoted to monitor and analyze vital signs with wearables [193–198], but to the knowledge of the authors, only a few recent publications are related explicitly to the development of IoT-based solutions [1,199].

Smart garments can also help patients during rehabilitation by assessing their daily activities [200]. In addition, certain authors proposed a Bluetooth-enabled smart garment based on Adafruit Flora for rehabilitating shoulders in physical therapy treatments [201], while others focused on knee osteoarthritis rehabilitation [202]. Moreover, haptic wearables (i.e., with sensory augmentation) can be used for the rehabilitation of patients with sensory impairments [203]. Furthermore, exoeskeletons and biomechanically assistive garments, which can be considered sorts of smart wearable, can help during rehabilitation processes to help walking or leaning to lift weights [204,205].

3.2.2. Vulnerable Groups: Baby and Elderly Monitoring

Dependent people have also been studied thoroughly in order to provide systems to monitor them. Thus, smart garments have been designed for babies and the elderly with the objective of tracking their movement, falls or general health. For instance, the authors of [206] describe a system for monitoring newborns through a smart garment that collects health information and transmits it to Bluetooth gateways in order to upload it to a remote server. A similar baby vest is detailed in [207],

although the proposed system is attached directly to a laptop running LabView instead of transmitting the signals wirelessly.

Another interesting work is described in [208], where a harness for preventing accidents in babies that crawl is presented. Thus, dangerous events are detected by the system, which uploads the information to an IoT platform that is able to warn remote care takers. Other authors focused on preventing infant death syndrome by monitoring the child's temperature, heart and breath rate [209].

Regarding the elderly, in the last years a lot of research has been performed due to the interest in topics related to aging. Thus, different authors have made use of smart garments to analyze diverse factors that influence the daily life of the elderly. For instance, the authors of [210] describe a smart shirt for monitoring the elderly that is able to collect physiological data (e.g., heart rate, respiration rate, underwear temperature) that can be transmitted through WiFi to a remote server where such data are processed and displayed. A similar approach is detailed in [211] but for collecting ECG and accelerometer data related to walking (in this case the information is transmitted through Bluetooth to a smartphone, which processes it and allows for performing emergency calls).

### 3.2.3. Sports and Wellness

In the case of sports, it is a sector that has traditionally been an early adopter of new textiles. The same has happened with smart clothing, which are really useful when training [212], performing certain activities [213] or when healing from sport injuries.

A good posture is essential both for sports and health, so its recognition has been studied through the use of intelligent garments. For instance, in [214] the authors present a smart shirt for monitoring posture that transmits the collected data to a PC. The system is not IoT-enabled, but it would be straightforward to send the received data to a remote cloud or processing service. Another posture-recognition vest is described in [215], which is able to stream real-time data through Bluetooth to a remote PC and detect relevant events.

Sedentariness is a dangerous habit that may harm wellness [216] and that has been analyzed by different researchers in order to prevent it through smart clothing. A recent example is described in [217], which presents a smart t-shirt that embeds physiological, ambient and motion sensors in order to detect sedentary lifestyle with the aid of machine learning techniques. Other authors detailed similar systems to detect physical activity, but based on a pair of modified glasses that are also able to estimate food intake [218].

In order to determine a level of well-being, the authors of [219] designed a smart sock based on the Lilypad Arduino that is able to collect sensor data and transmit them through WiFi to a remote database. Specifically, the smart garment collects data on the wearer's heart rate, heart rate variation, oxygen saturation, galvanic skin response or his/her temperature. With all such data, the system provides a score that estimates the user's well-being.

Finally, it is worth pointing out that the increasing and eventually ubiquitous use of wearables, as well as the constant streaming of bio-signals into an open blockchain-based system will gradually pave the way for collaborative and crowd-sourced wellness. The obtained bio-signals will enable greater transparency around outcomes of health treatments as well as new forms of diagnosis, clinical trials and personal care [220].

### 3.2.4. Industry, Defense and Public Safety

Smart wearables and garments can also be used in the harsh working industrial environments or in mission-critical scenarios [19], where they are able to measure bio-signal indicators such as heart rate, respiration, fatigue, stress, location or the presence of poisonous gases. Moreover, in such scenarios wearables may provide real-time feedback to the users, which can enhance its productivity and safety.

In many industries, it is also common work-related musculoskeletal and cardiovascular disorders, which are often due to excessive workload. To improve traditional risk assessments, which are usually performed via self-reports or observations and that have relatively low reliability, the authors of [221]

developed a smart vest with diverse sensors (i.e., ECG, Thoracic Electrical Bio-impedance (TEB), an Internal Measurement Unit (IMU)) that were connected through Bluetooth to an Android tablet where data were processed, stored and visualized.

The Defense and Public Safety sectors are already using wearable computers and smart clothing for access to positioning, sensor or health status data [222,223]. Moreover, in some countries the police and fire fighters have also been using head-mounted cameras for recording events. In the case of fire fighters, different researchers proposed systems to help them. For instance, in [224] the authors present a system for monitoring physiological parameters of a firefighter. The proposed system consists in a smart vest that exchanges information with a remote ZigBee coordinator that is attached to a PC where the physiological data are collected and stored. A very similar ZigBee-based system is presented in [225] while, in [110], diverse prototypes of smart garments (i.e., a smart T-shirt, a jacket and a pair of boots) are described. Such smart garments were conceived with the objective of remote monitoring the position, ambient and physiological status of emergency-disaster personnel.

### 3.2.5. Interaction with the Environment

Smart clothing can interact with certain IoT products, becoming part of such digital systems, like it is the case of the connected or autonomous car. For example, a smart garment that measures sleepiness can communicate its information to a central control system, which in turn can trigger a warning message or even take the appropriate actions (e.g., to decide to drive in autonomous mode or block the starter of the car to the driver).

Gestures can also be recognized in order to interact with the surrounding environment. For example, in [226] it is described a smart wristband based on a Lilypad Arduino that is able to detect gestures and then turn on/off and adjust the brightness of a home lighting. Other authors suggested that such an interaction with IoT systems may be automated by incorporating NFC tags into garments [227]. Furthermore, some researchers have also proposed to interact with robots through intelligent garments [228].

## 4. Market Opportunity

Over the last couple of years, a variety of smart wearables, including smartwatches, smart wristbands, smart glasses, smart garments and smart jewelry, have been launched around the world. Although, as it can be concluded after reading the previous section, smart clothing can be used in multiple applications, its global market is still in a nascent stage, with the major market traction for wearables being predominantly witnessed in North America and Europe [229,230]. However, increasing ICT spending, rising health concerns and more smartphone users are driving demand for smart wearables and clothing in other countries like China (e.g., 43% of urban Chinese consumers would buy wearable devices [231]). Consequently, major smart wearable and clothing players are focusing on offering technologically advanced products at low price points to increase their market globally. In the case of wearables, market prospects are very promising: wearable shipments are forecast to increase to US\$150 bn by 2026 from the estimated level of US\$30 bn in 2016. For instance, the AR technology market is expected to grow significantly in the next years together with the Virtual Reality (VR) technology market, creating a market of US\$80 bn in 2025 [232].

In addition, a BusinessWire report [230] found a growing global market for smart wearables that was projected to generate revenue worth \$41 bn by 2020 with a Compound Annual Growth Rate (CAGR) of 65% (CAGR represents the mean annual growth rate of an investment over a period of years). As it can be seen in Table 4, according to IDC (2018) Apple and Xiaomi are dominating the market, but certain investment will probably arrive from other companies that are not players in today's consumer electronics market. The introduction of innovative products is expected to influence significantly the consumer interest in the coming years.

**Table 4.** Top 5 worldwide wearable device companies, shipments (in millions), market share and year-over-year growth, Q2 2018. Source: IDC Worldwide Quarterly Wearables Tracker, 4 September 2018.

| Company | 2Q18 Shipments | 2Q18 Market Share | 2Q17 Shipments | 2Q17 Market Share | Year-over-Year Growth |
|---|---|---|---|---|---|
| 1. Apple | 4.7 | 17.0% | 3.4 | 13.0% | 38.4% |
| 2. Xiaomi | 4.2 | 15.1% | 3.5 | 13.3% | 19.8% |
| 3. Fitbit | 2.7 | 9.5% | 3.4 | 12.8% | −21.7% |
| 4. Huawei | 1.8 | 6.5% | 0.8 | 3.1% | 118.1% |
| 5. Garmin | 1.5 | 5.3% | 1.4 | 5.4% | 4.1% |
| Others | 13.0 | 46.6% | 13.8 | 52.4% | −6.2% |
| **Total** | **27.9** | **100.0%** | **26.4** | **100.0%** | **5.5%** |

A research report from Juniper Research forecasts that fitness garments and ear-based wearables will grow from 4.5 million shipped in 2018 to nearly 30 million in 2022, an increase of more than 550% [233]. It is also worth pointing out a forecast from IDTechEx [234] that projects that 3 bn sensors will be sold for wearable devices in 2025. One of the fastest growing segments is in stretch and pressure sensors, which, for instance, are essential for detecting motion of individual body parts. IDTechEx predicts that this segment will have a CAGR of 40% over the next 10 years. New chemical sensors will grow nearly as rapidly at a CAGR of 32%, optical sensors at a CAGR of 13%, and biopotential sensors at a CAGR of 10.8%. Even inertial motion sensors, the ones with the slowest growth in this forecast, will continue to increase at a CAGR of nearly 10%.

TechSci [235] projects that the demand for smart garments will continue to grow, driven particularly by athletes and diabetic patients. Other technologies presented as emerging are smart undergarments or inner-wear, smart tattoos and hologram projector-based smart wearables. Analysts believe that companies will increasingly focus on advanced technology, compatible and lightweight smart wearables. Alternatively, Gartner [236] also claims that smart garments will outstrip all other types of wearable fitness-tracking gadgets in terms of growth, going from nowhere to become the single largest category. According to the Ericsson ConsumerLab [237], within the next five years users are expected to wear multiple devices (i.e., electronic assistants, smart contact lenses, ear implants or even pills and chips under the skin) allowing them to connect to the Internet and to interact with their physical environment.

Regarding the textile business today, global textile, fashion and garment industry (including textile, clothing, footwear and luxury fashion) is currently worth nearly $3000 trillion [238]. The clothing industry produces 19 bn garments per year, 150-times the number of smartphones. The global market for smart fabrics is forecast to grow to around $4.08 bn by 2023 at a CAGR of 19.01% over 2018–2023 [239].

While current smart garments may be a small market share largely confined to sport, technology-enabled clothing could well be about to find a larger audience outside the fitness industry, disrupting the global textile market as we know it today.

*Competitive Environment and Impact on the Global Industry*

Unlike the computer or mobile industries, IoT-based wearables and garments are not yet dominated by established players. Nowadays, Small and Medium-sized Businesses (SMBs) of electronic components and systems (including organic and form-fitting electronics) are especially active in this field. Research and technology organizations are the main source of state-of-the-art innovations. Industrial players are actively building prototype systems and wearable solutions for areas like health, well-being and smart textiles, with an increasing potential to tap into future high volume markets and consolidate as a crucial part of the digital technology value chain for future products.

Manufacturers and Original Design Manufacturers (ODMs) are beginning to see the opportunity and currently separated supply chains are expected to merge. Teaming up with fashion designers with an eye towards mass production may quickly become mainstream. But, considering the confidential nature of the data and perceived level of risk [240], consumers are likely to choose only a few selected brands over others to commit to and invest in as guardians of their personal bio-signals [241]. Loyalty and brand name will play a big part in which platforms dominate the space. Gaining this highly desired access to biometrics will be one of the aims of the business players.

Beyond the explosion of sensor embedding in clothing, connected e-textiles will enable progressively the integration of various degrees of flexible displays (e.g., Flex9H [242] is a technology developed by Solip Tech Co., Ltd of Korea (Daejeon, Korea) which allows to roll up a display without worrying about breaking it), one day even possibly challenging the smartphone supremacy.

The clothing industry represents the largest potential distribution network for wearables. Consequently, it would be in retailers' best interest to learn how to market digitally enabled products, as well as master the high level of product complexity involved. Electronics retailers should also collaborate to market clothing-style form factors with their own set of intricacies, from body sizing to seasonal collections. Thus, retailers can leverage from advances in the printed electronics industry, which are going towards more flexible, thinner and high throughput processing. Textiles can be very inexpensive and printed electronics are getting a lot cheaper, so the integration of both technologies in a seamless manner can create an opportunity to make a durable and low cost platform.

Soreon Research [243] has analyzed more than 200 wearable projects, both from the start-up scene as well as from large corporations. For instance, it indicates that Google is looking for third parties to develop wearables on its Android Wear platform [244]. Apple is looking to become the go-to health hub for consumers through Healthbook [245]. Samsung, Intel and other tech giants are investing massive capital to ensure life bio-signals will soon be a reality and most tech-powered brands have also started to commit and partner. In other words, tech companies are all jumping in, eager to contribute to and profit from the biggest technological wave since mobile, hoping to protect from unwelcome disruptors, or to challenge, even dethrone, major tech brands.

Ralph Lauren [136], Polar [246], Under Armour [247] and others will be a force to be recognized within the wearable space provided they can develop core hardware and acquire digital product development know-how in a short time. In fact, the three previously mentioned companies have developed shirts that have sensors woven into their fabric and a transmitter module (typically positioned in the middle of the chest, at the back of the neck or in a side pocket) that exchanges data with a smartphone connected via Bluetooth.

Google and Levi's teamed up as Advanced Technology and Projects group (ATAP) through the project Jacquard [248] to make digitally connected clothing using a new kind of conductive yarn and woven multi-touch panels, so they can turn normal clothes into interactive devices [249]. ATAP has been working with textile experts in places like Japan to create a conductive yarn that can withstand the industrial weaving process and that also look good enough to make real clothes. ATAP's Jacquard is like a blank canvas for the fashion industry: designers can use it as any fabric, adding new layers of functionality to their designs without having to learn about electronics. Developers will be able to connect existing apps and services to Jacquard-enabled clothes and create new features specifically for the platform. ATAP is also developing custom connectors, electronic components, communication protocols and an ecosystem of simple applications and cloud services. The commuter trucker jacket is available for $350 on selected stores from September 2017 and it is able to stop or start music, to navigate with Google Maps, to take calls or to read incoming text messages just by swiping or tapping on the jacket sleeve. However, it must be noted that the jacket is designed to withstand up to 10 washes during its whole lifespan and the warranty coverage is only one year.

Other interesting initiatives are carried out by the American Department of Energy's Advanced Research Projects Agency (ARPA-E), which has been financing projects related to smart textiles with a budget of $30 million within the Delivering Efficient Local Thermal Amenities (DELTA)

program. For example, a project called Adaptive Textiles Technology with Active Cooling and Heating (ATTACH) [250], which received a $2.6 million ARPA-E award, aims for automatically increasing or decreasing the insulation properties (i.e., porosity and thickness) of textiles in case the ambient temperature decreases or increases, respectively. Such an adaptive mechanism would be purely passive, so it would consume no power. In addition to that, the ARPA-E team plans to add integrated heating and cooling elements by using printed thermoelectric devices and batteries, and innovative ways of powering the active components. Among these are a flexible biofuel cells that use sweat as a fuel source, and printable flexible low-cost batteries.

Finally, it is worth indicating that mergers and acquisitions are leading to the consolidation of business strategic alliances: Microsoft paid between $100 and $150 million to Osterhout Design [251] for its IP portfolio related to high-tech wearables for industrial and defense applications; mCube [252] received $37 million funding to develop sub-miniature Micro-Electro-Mechanical Systems (MEMS) sensors for wearables; SiliconLabs acquired Bluegiga Technologies Oy [253] for $61 million, to strengthen their protocol expertise in Bluetooth Smart; and BAE Systems partnered with Intelligent Textiles Limited to create Broadsword® Spine® [254], an invisible power and data network built into clothing.

## 5. Main Challenges and Technical Limitations for a Broader Adoption

The full deployment of smart garments in new application areas requires further advances to increase functionality, to enable adaptability and to reduce energy consumption [255]. A shift will occur from today's single-function wearables (e.g., rigid electronics, peripheral mobile phones) towards the next-generation of IoT-based wearables and garments, whose main challenges can be summarized as follows:

- Garments need to be comfortable and flexible.
- Garments need to reach higher Technology Readiness Level (TRL) [256].
- Unlike smartphones, the target audience wears multiple clothing items, not just one high-end item. This means that this is really a multi-billion-piece technology market. It is probably the only one that can match and then overtake the volumes of the smartphone industry [47]. But therein lays the problem: selling multiple, everyday items, is a very different proposition to the current approach of the consumer electronics and technology industry. When it does take off, it will be very different to today's industry. Whereas the average consumer has a choice of a limited number, maybe a hundred different smartphones, the smart clothing market will provide them with a greater choice of clothing, each in a variety of different sizes. In addition, clothing often sells on individuality (i.e., many people usually do not generally like to meet others wearing the same clothes as themselves [257]).
- Made to measure, as a new form of tailoring, is being slowly brought into the mainstream by automated measurement and cutting, companies like True&Co are claiming the fact that they are using fit data from over five million women to design bras [258]. These techniques are still in their infancy. But sizing is a major challenge for smart clothing. Depending on the sensors that smart garments embed, they need to be a moderately good fit if the sensor is going to stay in contact with the body. That implies a degree of tailoring or made to measure that will keep prices high for the foreseeable future.
- The cost of smart fabric is another concern, but from an implementation standpoint, the need of the product drives the demand and that sets the market price. There are a lot of factors beyond just what it costs to price these items. Still, if a business desire is to make a ubiquitous platform that everyone in our community can access, it really needs to be cheaper than the wearable electronics currently available on the marker. At some point, made-to-measure will descend to a price point that brings it to volume.

- Stores are popular because shoppers enjoy the diversity of choice, product trial, and the social aspect of buying clothes [259]. Retailers will have to work out how the changing room will look like and how to promote the features of smart clothes.
- Smart environments rely on a ensuring high-security against cyber threats [260–262], and on the constant availability of sensor and actuator devices, whose power consumption is a concern due to the large number of sensor nodes to be deployed. IoT devices require high-security lightweight protocols [263] and cipher suites [264] that optimize the use of resources and the energy consumption.
- New IoT architectures will replace current cloud-based systems in certain scenarios like smart health where latency and communications have to be minimized to react fast to events. For example, the fog computing paradigm have arisen recently by moving the capabilities of the cloud towards the edge of the network [265–267].
- Charging and battery life still represent a research problem [268,269].
- Durability against deformation, and therefore ensuring a long lifetime and sustainable performance, is another of the biggest challenges facing smart textiles and sensors which use clothing or fabric as a platform for integrating electronics [270,271].
- The influence of washing processes, temperature, sweat, moisture, mechanical impacts, repeated bending and compression, light (especially sunlight) should be carefully considered.
- Retail businesses will have to determine how far they should go with biometrics and which are their responsibilities. The widespread of IoT smart garment technologies will raise data security concerns and privacy issues over the right to access data generated by millions of clothes. The companies that produce the garments and provide online accounts for tracking those data will have access to them and they will be able to sell such an information to marketing companies, insurance companies or healthcare providers. People will perhaps feel they will need to gain more control over where their data go and who is getting access to see them.
- Strategic alliances with technological partners will be needed to overcome many technical challenges that face smart clothing. There is still plenty of research to be done, especially in the areas of battery power, energy harvesting and hardware miniaturization [272] before the Internet of Smart Clothing hits the mainstream.

## 6. Conclusions

This article reviewed the past, present and future of smart wearable and clothing, providing a holistic approach on the topic. The main types of smart wearables were distinguished and the most relevant subsystems of a smart garment were described, as well as their communications architecture. In addition, the most relevant examples of smart clothing applications were detailed, showing the potential of IoT-enabled smart garments. Furthermore, the market perspectives and opportunities were analyzed in order to emphasize the potential of the next generation of smart clothing. Finally, the main challenges and recommendations for the deployment of the smart garment industry were enumerated. To sum up, this article provided guidelines for future IoT smart garment designers and developers with the objective of making reality the concept of the Internet of Smart Clothing.

**Author Contributions:** T.M.F.-C. and P.F.-L. contributed to the overall study design, data collection and analysis, and writing of the manuscript.

**Funding:** This work has been funded by the Xunta de Galicia (ED431C 2016-045, ED341D R2016/012, ED431G/01), the Agencia Estatal de Investigación of Spain (TEC2013-47141-C4-1-R, TEC2015-69648-REDC, TEC2016-75067-C4-1-R) and ERDF funds of the EU (AEI/FEDER, UE).

**Conflicts of Interest:** The authors declare no conflict of interest.

## Abbreviations

The following abbreviations are used in this manuscript:

| | |
|---|---|
| AR | Augmented Reality |
| ASIC | Application-Specific Integrated Circuit |
| BAN | Body Area Network |
| BLE | Bluetooth Low Energy |
| CAGR | Compound Annual Growth Rate |
| CBRNE | Chemical, Biological, Radiological, Nuclear, and Explosives |
| COTS | Commercial Off-The-Shelf |
| CPS | Cyber-Physical System |
| CPU | Central Processing Unit |
| FPGA | Field-Programmable Gate Array |
| GNSS | Global Navigation Satellite Systems |
| GPS | Global Positioning System |
| IoT | Internet of Things |
| LAN | Local Area Network |
| LCD | Liquid-Crystal Display |
| LED | Light-Emitting Diodes |
| LDR | Light-Dependent Resistor |
| LoRaWAN | Long-Range Wide Area Network |
| MEMs | Micro-Electro-Mechanical Systems |
| NAS | Network-Attached Storage System |
| NFC | Near Field Communication |
| ODMs | Original Design Manufacturers |
| OLED | Organic Light-Emitting Display |
| PAN | Personal Area Network |
| RFID | Radio-Frequency Identification |
| RSS | Received Signal Strength |
| RSSI | Received Signal Strength Indicator |
| SoC | System-on-Chip |
| SMBs | Small and Medium-sized Businesses |
| TRL | Technology Readiness Level |
| UV | Ultraviolet |
| UWB | Ultra-Wide Band |
| VR | Virtual Reality |
| WAN | Wide Area Network |
| Wi-Fi | Wireless Fidelity |
| WPAN | Wireless Personal Area Network |
| WSN | Wireless Sensor Networks |

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
