# Peer review of "Towards The Internet of Smart Clothing: A Review on IoT Wearables and Garments for Creating Intelligent Connected E-Textiles"

_electronics, doi:10.3390/electronics7120405_

Round 1

Reviewer 1 Report

This manuscript reviewed the main requirements for developing smart IoT-enabled garments and shows smart clothing potential impact on business models in the medium-term. It presented the main types and components of a smart wearable garments. Specially, it gave the commercial and academic applications in smart clothing. I considered that it was well written and organized. However, some small details can be added to make this manuscript more complete in the review.

1)   In page 4, concerning about the classification of smart wearables, the European Union has published a Technique Report CEN/TR 16298:2011. It was an important guideline for the researchers. The author should present this report.

2)   In page 8, line 235-238 for Vita sign rates section, some references such as, https://onlinelibrary.wiley.com/doi/abs/10.1002/admt.201700309could be added.

3)   In page 13, display subsystem, the optical fibre and electrochromic display have been well developed in the smart textile applications. Authors could add some references in these fields.

4)   The interconnection subsystem is a crucial key for the reliability and washability for smart textile. In the same time, it decides the possibility of miniaturization and wearability. The author should present the short review in this aspect.

I think that this manuscript can be considered as accepted after the author follows the comments.

Author Response

Dear Sir/Madam,

Please find attached our detailed responses to the comments. 

Regards.

Reviewer 2 Report

Overview

This manuscript provides a review of electronic textiles and wearable devices, with a focus on IoT enabled devices. The manuscript covers three core areas; the components that make up a smart garment, a review of current and future devices, and an analysis of future market opportunities. This work would be of interest to the growing wearables and e-textiles communities, and its holistic approach would make it particularly useful for people wanting an overview of the area. The overview on Market Opportunities at the start of section 4 is particular interesting. Unfortunately, there are issues with the manuscripts structure and content, which have been detailed below.

General comments

1.      The manuscript is listed as an article but is really a review, as indicated by the title. This should be updated.
2.      The current narrative is somewhat unclear. This work is really a review and discussion of wearable computing with a focus on E-textiles, but at points the narrative is confusing as the review moves between wearable and E-textiles; this is very apparent in Section 3.1. Further, some areas are well described, while others are skimmed over. This may give the reader an unrepresentative view of the actual state of the field. While the paper states that it is a 'comprehensive analysis of the evolution of smart wearables and garments, and their main characteristics and applications', reading this manuscript as it is currently written, this is not comprehensive. The evolution of smart wearables and garments in particular is very short.
3.       This review would be a lot clearer if sections 2 and 3 were restructured. For example, both sections 2.4.2 and section 3 discuss sensors, which ultimately makes both sections incomplete. The authors may wish to consider re-writing these sections based on function.
4.      This review is very long and the authors should be more economic with language. There are areas of text without supporting citations that do not add a lot to the overall message, for example, I am not certain what lines 510 – 524 add to the narrative of this work. The introduction to this section could be reduced to a sentence (or simply start at line 525). Similarly, I think that the overall introduction could start on line 41. This would help with the structure of the piece, and make it easier for the reader.
5.      There are a number of times (some highlighted in the specific comments section) where a citation is needed to back-up statements.
6.      More references are needed in section ‘2.4.2. Sensing subsystem’. Many sensors are discussed without a reference and it is unclear to the reader what has been done or what could be done. At least one reference should be given to each of the sensor types mentioned. It would also be beneficial to draw the reader’s attention to comprehensive reviews into this area such as 'Smart fabric sensors and e-textile technologies: a review' and 'Wearable electronics and smart textiles: a critical review'. Section ‘2.4.3. Actuation subsystem’ has a similar problem and requires more citations.
7.       The language in section ‘2.4.4. Control subsystem’ needs to be revised. There are a number of grammatical errors. The way that this sub-section has been written makes the authors message unclear at points. I have included a couple of specific comments later in this review.
8.      There are minor grammatical errors throughout this manuscript. Some have been listed in the ‘Specific Comments’ section.
9.      Section 2.4.7 should better discuss the battery technologies and better explain why they are not suitable for smart clothing. This would possibly be improved by including a summary table. This whole section is poorly cited and requires expansion and clarification. Line 359/360 states that ‘There are two main energy sources that may be harvested by smart garments’, which is not true. There is significant work in the literature where solar has been used (solar is discussed on line 363). The section does not discuss any of the significant body of work into wearable energy solutions (specifically energy storage solutions).
10.  Section 3.1 does not necessarily represent the ‘main commercial applications’ in this sector. I did not check every single reference but many of the citations that I checked are not for commercially available products (i.e. 120, 129), or do not represent a proper commercial application (i.e. 137 which is really an artistic piece). Beyond this, this list is missing a number of commercially available products in this sector. This shows a lack of understanding by the authors with regards to this field and recent and near-future commercial developments.
11.  Further critical analysis of the applications discussed in section 3 is required.

Specific comments

Line 19/20: The keyword electronic textiles (as opposed to e-textiles on its own) would likely increase visibility.

Line 25: I do not necessarily agree with this statement. The concept of smart textiles encompasses textiles that can interact with the environment/user, so can include reactive inks. ‘Smart clothes can be created by embedding smart wearables into garments.’ would be more accurate.

Line 27: These statements should be supported with references.

Line 28: These applications should be supported with references.

Line 31: Why is this a so-called wearable?

Line 33: The statement should be supported by a reference.

Line 34: This really needs to be expanded to clarify what you mean.

Line 45: '...and smart textiles' should have a reference. To my understanding there aren't any examples of smart textiles coupled with 5G.

Line 48/49: '...have the potential to transform society.' This needs to be strengthened. It is not really clear how this will happen from the text.

Line 71: I would consider revising this title to be more descriptive.

Line 72: This title should also be revised. This is really a brief overview of early wearable computing and the title should reflect this.

Line 92: This does not make sense and must be re-worded.

Line 111: I do not think the use of the comma is correct.

Line 135-137: Where has this distinction come from? I have never seen e-textiles separated into these categories, level of integration within the textile is a far more common method of distinction.

Line 137: This does not make sense and needs revising.

Line 138/139: I understand what you are trying to say, but this is not clearly communicated. This sentence needs to be revised/expanded.

Line 145: This should have a citation.

Line 173: The placement of Figure 1 and 2 should be revised; currently they appear halfway through a list.

Line 257: Optical fibres should also be included here.

Line 264/265: This is a major area of E-textile research and has led to a number of successful products. Historically this is the main area of E-textile development. This area should be expanded upon as the way this is currently written undersells the importance of this application to the reader.

Line 272/273: The statement on FPGAs should be clarified.

Line 276: The use of the term ‘extremely powerful’ needs to be clarified.

Line 316: This paragraph should really be expanded. The uses of RFID as a communication subsystem is extremely limited, yet you have covered this in detail and not really discussed communication technologies that would (potentially) allow connection to the internet or for off-device processing.

Line 338: I disagree with this statement, power is essential for electronic textiles. This does not necessarily need to be batteries (supercapacitors for example).

Line 351: You state that Li-Ion batteries are bulky and add weight to the garment, but do not say the same for bulkier and heavier AA batteries a few lines earlier. You should clarify (or quantify) what you mean regarding the bulk and weight.

Line 372/373: This is typically the case, but needs clarification and/or a citation (raw data can be transmitted to another device for processing).

Line 374-376: Please revise bracket use.

Line 378/379: ‘The connection with the back-end is frequently performed via smartphones,’ this requires a citation.

Line 385: Why untrusted?

Line 418: The wording of this sentence needs to be revised.

Line 430/431: A citation is required.

Line 432: The current placement of the figure breaks a sentence. It would be better if the figure was moved.

Line 445: This statement should be supported by some figures/a reference.

Line 512: ECG and EEG should also be written out.

Lines 538-540: Additional discussion and citations are needed regarding the other vital signs that can be monitored. There is a wealth of literature out on these.

Line 667/668: This requires a citation.

Line 686-690: You need to explain why ‘consumers are likely to choose only one or a few selected brands over others to commit to and invest in as guardians of their personal bio-signals’ with supporting references to show that this is a likely market trend. Otherwise this is conjecture.

Line 711: This should read 'be a force'.

Line 733: The cost needs to be stated to support this (I realise that this varies depending on application, but some indication should be given).

Line 779/780: This requires a citation.

Line 791: Why is durability a problem? This needs to be discussed.

Author Response

(The authors gave the same response as above.)

Round 2

Reviewer 2 Report

The authors have significantly modified the manuscript in-line with the reviewer comments. I appreciate the fact that the authors have clearly and carefully reviewed each comment, and overall I am happy with the changes made. As I said in my original review, I believe that this work will be of interest to the community. The revisions to how the paper is written (in particular the introduction) makes the narrative and focus of the paper a lot clearer. However, there are still a few minor corrections that I believe are necessary before this can be published.

General comments

1) While reading the manuscript I noticed some (minor) grammatical errors. I have highlighted any errors that I found in the specific comments section, but I may have missed some things. The manuscript should be checked again for grammatical mistake.

Specific comments

Line 28: ‘textile is essentially’ should be ‘textiles are essentially’

Line 107: Should read ‘in Artic environments’.

Line 140: I do not think that these references adequately back-up this statement. The presentation is no long available, and, while I did not read the entire PhD thesis (ref 47), I don’t think that this backs up the argument that there are five type of levels of integration for e-textiles. As I have said in my previous comment, I have never seen the level of integration described in this way. Page 31 of the cited thesis discusses the generally accepted view of integration. I think that showing different types of attachment of electronics onto textiles is important but I think that the paragraph (lines 138 to 141) and figure need to make it clearer that there are (generally speaking) two/three levels of integration, and then multiple methods of attaching electronics (with attached electronics being one level of integration).

Line 147: Figures 1 and 2 break a sentence. Their placement should be revised.

Line 278: I think that the use of the word powerful should be revised: ‘has significant processing power’ may be better.

Line 293: ‘what’ should be ‘which’.

Line 300/302: I think that the word ‘computational’/’computationally’ should be added in front of power/powerful. While I understand what you mean, this may be confusing to a causal reader.

Line 589: ‘For instance, in [207] it is described a system…’ should read ‘For instance, in [207] a system is described…’

Line 594: ‘where it is proposed a sort of harness’ should be ‘where a sort of harness is proposed’, however I would consider revising the word choice for ‘sort of’.

Line 597: ‘preventing infant death syndrome by monitoring child’s temperature’ should be ‘preventing infant death syndrome by monitoring the child’s temperature’.

Line 600: ‘in [211] it is described a smart shirt’ should be ‘ [211] described a smart shirt’

Line 677: This is not grammatically correct. I would consider changing it to… ‘The AR technology market is expected to grow significantly…’

Author Response

(The authors gave the same response as above.)
